# Endemic Plants Can Be Resources for Mountain Agro-Ecosystems: The Case of *Sanguisorba dodecandra* Moretti

**Luca Giupponi** [1,2] , **Valeria Leoni** [1,*] , **Carla Gianoncelli** [3] , **Alberto Tamburini** [2] **and Annamaria Giorgi** [1,2]

1 Centre of Applied Studies for the Sustainable Management and Protection of Mountain Areas (CRC Ge.S.Di.Mont.), University of Milan, Via Morino 8, 25048 Edolo, Italy; luca.giupponi@unimi.it (L.G.); anna.giorgi@unimi.it (A.G.)
2 Department of Agricultural and Environmental Sciences—Production, Landscape, Agroenergy (DISAA), University of Milan, Via Celoria 2, 20133 Milan, Italy; alberto.tamburini@unimi.it
3 Fondazione Fojanini di Studi Superiori, Via Valeriana 32, 23100 Sondrio, Italy; cgianoncelli@fondazionefojanini.it
* Correspondence: valeria.leoni@unimi.it

**Abstract:** *Sanguisorba dodecandra* Moretti is an endemic plant of the Alps of the Lombardy region (Northern Italy). Differently from most endemic species, this plant grows in diverse environments, and it is often very abundant and a distinctive element of some mountain and sub-alpine agro-ecosystems. The ecological features and the role of this species in some mountain agricultural activities are poorly investigated. This article shows the results of a synecological analysis of *S. dodecandra* and the evaluation of its functional strategy. Furthermore, its forage value was investigated and melissopalynological analysis was used to characterize the honey produced in an area where this species grows. The ecological analysis defined this plant as euriecious and ruderal/competitive-ruderal strategist. Bromatological analysis showed a good forage value, confirming the ethnobotanical knowledge concerning this species. In fact, it has good protein content (12.92 ± 1.89%) and non-fiber carbohydrates (47.12 ± 3.62%) in pre-flowering. *S. dodecandra* pollen was identified as a "frequent pollen" in the honey, showing that this plant is attractive to honeybees. This research allowed a deeper knowledge of *S. dodecandra* ecology and showed that this species is a resource for traditional and sustainable agricultural activities of the Lombardy Alps such as pastoralism and beekeeping.

**Keywords:** mountain plant resource; endemic plant; mountain agro-ecosystems; forage value; honey; functional strategy; synecology; pollen; honeybees; Orobic Alps

## 1. Introduction

Alps have a floristic heritage of 4.500 tracheophytes, of which 9% are endemic to the Alpine Mountain Range [1,2], and so they represent a valuable element of this habitat. Part of the Southern Alps, such as the biogeographic unit "Orobic Alps and Lombardy pre-Alps" [3], are hotspots of endemic species for historic and evolutionary circumstances [1,3,4]. Endemic species from the Orobic Alps and Lombardy pre-Alps make up approximately 6% of the native flora (including archaeophytes) [5]. Most of them are basiphilous, but there are also some that are acidophilous [3,5]. Endemic plants are generally very rare although some specific cases are surprisingly abundant in the environment where they grow, becoming an important and exclusive resource for agri-pastoral activities in mountain territories. One of these cases is *Sanguisorba dodecandra* Moretti.

*Sanguisorba dodecandra* is a perennial herbaceous species (hemicriptophyte) of the Rosaceae family, endemic in the Alps of Lombardy (Figure 1) [6]. This plant grows almost exclusively in the Orobic Alps (Bergamo Province and Sondrio Province), and only a few stations of the Rhaetian Alps are known (Valtellina, Sondrio province) [5]. Some of these growing sites are the result of introductions realized in 1925 and 1935 with the aim of investigating the ecology of the species and its capacity to spread [7]. The current

distribution of this species is of recent origin, since it grows in areas covered by glaciers during Würm glaciation [8].

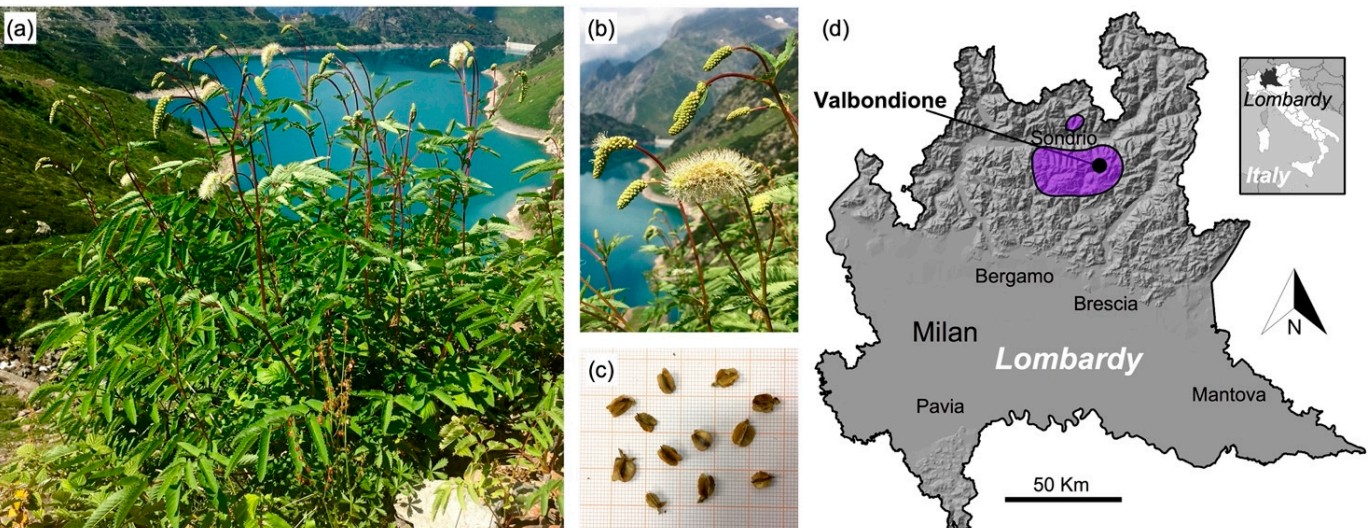

**Figure 1.** *Sanguisorba dodecandra*: Whole plant in its growing environment (Valbondione—BG) (**a**), inflorescence (**b**), fruits (achenes) (**c**). Distribution range of *S. dodecandra*, according to [5], highlighted in purple (**d**).

*S. dodecandra* is an acidophilous species with an uncommon abundance for an endemic species [3], and it is often dominant in the plant communities in which it grows. This species is considered "Near Threatened—NT" according to the IUCN Red List of threatened species [9]. *S. dodecandra* grows in disparate environments all characterized by high edaphic and/or air humidity: In avalanches gullies, nearby alpine streams and lakes, over dripping rocks/debris, and humid meadows and pastures [10–12], in an elevation range between 1000 and 2300 m a.s.l. [5]. However, different studies stated that this species can find its optimum in the *Cirsio-Sanguisorbetum dodecandrae* (*Adenostylion alliariae* alliance, *Adenostyletalia* order, *Mulgedio alpini-Aconitetea variegati* phytosociological class), especially concerning Valtellina [10,12,13].

While detailed studies of plant communities of *S. dodecandra* were performed in Valtellina, both in the Rhaetian and Orobic Alps of Sondrio province [12,13], none of the analogous investigations were performed in the Orobic Alps of Bergamo province, where the species spread over a wider area [5,11,12,14–16]. Moreover, we did not find any data on synecological features of this species, clarifying its climatic and/or soil requirements. Some ecological features are not known as well, such as the evaluation of the competitor, stress-tolerator, ruderal (CSR) functional strategy [17–19]. CSR analysis is based on Grime [17–19] theory. According to Grime theory, plants, to survive, can adopt different strategies, meaning that they develop an adaptive response towards competition (biotic limitations to biomass production), stress (abiotic limitations to productivity), and disturbance (biomass destruction). Grime theory can be applied to the ecological characterization of wild and endemic plants [20–22] but also to crops and traditional cultivars (landraces) [23]. A database on the CSR plant strategy of many species was built [24], but *S. dodecandra* is not present in this database. However, it is indicated as a C/CSR (competitive/competitive-stress-tolerator-ruderal) species in [25] without any details on the number of samples examined, the area, and the sampling period. Moreover, in [26], the competitive/stress-tolerator ("css") strategy was attributed to this plant by merely an empirical observation.

*S. dodecandra* has other features poorly or not at all investigated, which could be interesting for sustainable development and/or the valorization of agri-food value chains in the mountain territories where it grows. The first is related to its use as forage for

mountain transhumance activities, while the other is related to its potential as a nectar source for beekeeping activities.

Dairy production of traditional cheese and honey are strongly linked to a territory are the representation of the different alpine environments, from the meadows at the bottom of the valleys to the high-altitude grasslands, and they can then represent an opportunity for sustainable development for mountain marginal areas through added-value and unique-quality products [27]. Beekeeping can help economically vulnerable communities, such as the ones of marginal mountain territories, through honey production, as well as other economic benefits such as pollination services, agriculture, and forestry usage [28]. Honey organoleptic features are influenced by its botanical origin, which is then fundamental to defining this product [29,30]. Likewise, the different predominant plants of grasslands influenced the chemical and sensory characteristics of cheeses made with milk from animals grazing in highland pastures [31]. The habitat of many alpine plants is undergoing changes due to global warming [32], and the Alps are experiencing a process of abandonment [33]. Therefore, agricultural practices were progressively neglected [34], with forest expansion in areas once occupied by mountain grasslands [35], and activity exploiting these resources in a sustainable way should be encouraged, supported by a strong scientific backing on the features of these resources.

Considering the forage value of this plant, no scientific (bromatological) data exist to date, but there is diffused common knowledge related to this plant by farmers and shepherds. Doctor Giuseppe Filippo Massara (1792–1839), who found and described *S. dodecandra* for the first time in 1829 in Valtellina [36], stated that it was mown by farmers and considered an excellent fodder plant, ensuring the production of excellent cheese and butter [15]. Mountain shepherds considered this species of high palatability for cows until today (direct testimony).

No specific investigations were performed on the possible value of this species to produce mountain wildflower honey, one of the excellences of the meadows and pastures of the Alps [37]. It is known that flowers of *S. dodecandra* have nectaries at their base [10] and are organized in eye-catching blossoms (capitulum) (Figure 1a,b) suggesting an entomophilous pollination. The pollinators are, however, not known, and it is undetermined if the honeybee (*Apis mellifera*) is among them.

In this study, therefore, we aimed to understand the ecological features of *S. dodecandra*, which are poorly known. In this context, its role in some mountain agricultural activities such as pastoralism and honey production was investigated. This was achieved by the floristic and ecological features of the plant communities of *S. dodecandra* in three sampling areas of Valbondione municipality (Orobic Alps of Bergamo province), at different elevations. Furthermore, the CSR functional strategy was evaluated considering the three phenological stages of the plant: Pre-flowering, flowering, and post-flowering. Finally, the bromatological analysis of samples of *S. dodecandra* collected in the three phenological stages and the melissopalynological analysis of one sample of honey produced in Valbondione were carried out in order to evaluate if this species can be a resource for the agro-ecosystems where it grows.

## 2. Materials and Methods

### 2.1. Sampling Areas

The data of plant communities of *S. dodecandra* and the plant material used to perform functional strategy analysis and bromatological analysis were collected in three sampling areas of the Orobic Alps (Bergamo province, Lombardy region, Italy) located in the Valbondione municipality (Figure 1, Table 1) and less than 5 km away from each other. The sampling areas are located at different elevation/belts (A, mountain belt; B, high mountain belt; C, sub-alpine belt above the treeline) where the coverage of *S. dodecandra* is greater than 75%, and therefore uniform and physiognomically similar to the point of view of vegetation. All the sampling areas are included in the Orobie Bergamasche Regional Park.

**Table 1.** Sampling areas.

| Sampling Area | Municipality | Latitude N | Longitude E | Elevation (m a.s.l.) | Slope (°) | Aspect (°) |
|---|---|---|---|---|---|---|
| A | Valbondione (BG) | 46°03′02″ | 10°01′40″ | 1215 | 36 | 250 |
| B | Valbondione (BG) | 46°03′08″ | 10°00′45″ | 1610 | 32 | 186 |
| C | Valbondione (BG) | 46°04′06″ | 10°03′51″ | 2000 | 41 | 165 |

In these three sampling areas, *S. dodecandra* grows on ranker soils (immature acid soils with an AC profile) and/or screes of the Collio Formation (Lower Permian), which consists mainly of acid sandstones and siltstones. These sampling areas belong to the South Orobic geobotanical district [38] of the Central and Eastern Alps ecoregional section (Alpine Province; Temperate Division) [39] and lie in the orotemperate ultrahyperhumid bioclimatic belt of the temperate oceanic bioclimate [40]. The average annual precipitation of Valbondione is greater than 1700 mm, with two maximums in the equinoctial periods and one in August due to summer storms [5].

*S. dodecandra* blooming corresponds to the period of production of mountain wild-flower honey, occurring in June–July–August. One nomadic beekeeper ("Rossini e Serafini" farm) was found to use the foraging area corresponding to the sampling areas, allocated within a distance lesser than five kilometers, for wildflower honey production. The honey sample (0.5 kg) used for the melissopalynological analysis was collected after decanting, in October 2020. This honey was produced in the summer months of 2020 in Valbondione in a production area located at 1000 m a.s.l. The beekeeper, before the melissopalynological analysis, declared that this honey resulted as more similar to "linden honey" than to wildflower honey, based on its color and flavor.

### 2.2. Vegetation Survey and Synecology

In each sampling area (A, B, and C) (Table 1), three phytosociological relevés were carried out in order to collect the data on the plant communities of *S. dodecandra*. The phytosociological relevés were performed in July 2020 using Braun–Blanquet's method [41] over an area of 25 m² (5 × 5 m). "Flora d'Italia" [6] were used for the identification of the plants (tracheophytes), and the conventional Braun–Blanquet abundance/dominance scale was used to evaluate plant coverage (r, rare species in the relevé; +, cover <1%; 1, cover 1–5%; 2, cover >5–25%; 3, cover >25–50%; 4, cover >50–75%; 5, cover >75–100%). The data of the relevés were arranged in a matrix (relevés x species) where abundance/dominance indexes were converted into the percentage of coverage as proposed by Giupponi and Leoni in [42] (r, 0.01%; +, 0.5%; 1, 3.0%; 2, 15.0%; 3, 37.5%; 4, 62.5%; 5, 87.5%) to perform numerical and statistical analyses. A hierarchical clustering analysis heatmap and principal component analysis (PCA) were performed to evaluate the floristic similarity of the relevés and show any differences among the sampling areas. Cluster analysis was performed using the unweighted pair group method with the arithmetic mean (UPGMA) and the Euclidean distance coefficient [42].

Ecological indices of Landolt et al. [26] were used to carry out the synecological analysis considering the presence/absence of the species: The index of temperature (T); continentality (K); light intensity (L); soil moisture (F); substrate reaction (R); nutrients (N); humus (H); and aeration (D). Synecological data were visualized using boxplots and analyzed using one-way ANOVA, followed by the Tukey post-hoc test in order to show the ecological indexes that are significantly different ($p < 0.05$ and $p < 0.01$) among the sampling areas. Statistical analyses were performed using R 3.5.2 software [43] and the "vegan" package, while the scientific names of the plants are in accordance with [6].

### 2.3. Functional Strategy Analysis

CSR functional strategy of *S. dodecandra* was performed according to the method proposed by [25]. Twenty fully expanded leaves of this species were collected in each sampling area (Table 1) considering three phenological stages: Pre-flowering (I), flowering (II), and

post-flowering (III). The leaf samples were collected from different plant individuals. All 180 samples collected (20 leaves × 3 study areas × 3 phenological stages) were wrapped in moist paper and put in a refrigerator at 4 °C for one night. Leaf fresh weight was measured from these saturated leaves using an analytical weight scale (Precisa XB 220A). The leaves were digitized with a digital scanner, and their leaf area was calculated using ImageJ [44]. Leaf dry weight was measured after oven drying at 105 °C for 24 h. CSR values and functional strategy were determined using the 'StrateFy' spreadsheet [25]. Finally, CSR values were plotted in the CSR ternary graph using the 'ggplot2' package [45] of R, and a one-way ANOVA followed by the Tukey post-hoc test was performed considering C, S, and R values as dependent variables and phenological stages as independent variables.

### 2.4. Bromatological Analysis

To evaluate the nutritional-bromatological features of *S. dodecandra*, 5 kg of fresh plants (above ground organs) were collected in the sampling areas (Table 1) at different phenological stages: Pre-flowering (I), flowering (II), and post-flowering (III). The 9 forage samples collected (3 phenological stages × 3 sampling areas) were grounded by a rotor mill (FRITSCH Variable Speed Rotor Mill Pulverisette 14) before carrying out the bromatological analysis. The samples, fresh and after being dried in an oven (MPM Instruments M120-VF) at 105 °C for 12 h, were weighed to calculate the dry matter content (DM). The ash content (AC) was calculated by incinerating the samples on a flame and then in a muffle oven at 550 °C for 5 h. The total protein content (PC) was calculated following the Kjeldahl method using an automatic analyzer (Kjeltec Auto 1030). The neutral detergent fiber (NDF) and the acid detergent fiber (ADF) were determined for each grass sample using the Van Soest method and an automatic analyzer (ANKOM 220). The Van Soest method was used to calculate the amount of acid detergent lignin (ADL), while the content of non-fiber carbohydrate (NFC) was calculated as follows: NFC = 100−AC−PC−NDF−ether extract.

The Soxhlet method was used to calculate the ether extract (soluble substances and fat in ether). Each analysis was performed in duplicate, and all bromatological parameters were expressed as a percentage of dry weight (DW), except for DM, which was expressed as a percentage of fresh weight (FW).

### 2.5. Melissopalynological Analysis

Melissopalynological analysis of the honey sample was performed according to the techniques proposed by the International Commission for Bee Botany (ICBB) [30]. The following method was used for slide preparation: 12 g of honey was weighted in a 50 mL test tube with conical bottom and dissolved with 40 mL of distilled water. The solution was centrifuged (15 min at 3000 rpm), and then the supernatant was separated from the sediment by aspiration. To remove the last sugars present, the sediments were dissolved again in 10 mL of distilled water and centrifuged again (5 min at 3000 rpm), and the supernatant was eliminated by pouring. The sediment was dispersed again and transferred to a glass slide and finally distributed over an area of approximately 1 cm$^2$. The glass slide was left to dry at 40 °C. Once dry, it was included in a drop of glycerol gelatine and covered with a coverslip. Pollen grain counting was performed according to the method described in [46]. Examination under the microscope was carried out at the magnification that was most suitable for identifying the various elements in the sediment (400 to 1000x). If the sediment contained a high percentage of overrepresented pollen, a second count excluding the over-represented pollen was performed in order to determine the relative abundance of other pollen types more precisely. The pollen types present in the honey samples were identified, counted, and classified, according to their percentages [47].

## 3. Results

### 3.1. Plant Community and Synecology

During the study, 99 tracheophytes were identified (Table S1). Most of them are herbaceous perennial plants (hemicryptophytes) commonly found in the Alps [2]. The following

10 species appear in more than 55% of the relevés: *Sanguisorba dodecandra*, *Brachypodium sylvaticum*, *Chaerophyllum villarsii*, *Fragaria vesca*, *Hypericum maculatum*, *Rumex scutatus*, *Achillea millefolium*, *Carduus defloratus* subsp. *rhaeticus*, *Dactylis glomerata*, and *Rubus idaeus*. Most of them are diagnostic species of the *Mulgedio alpini-Aconitetea variegati* phytosociological class (and subordinate units) that encompasses communities of megaforbs and tall grasses of eutrophic and humid soils, rich in organic matter [48,49].

Figure 2 shows the floristic difference among the plant communities of the three sampling areas. The presence of groups of species with different thermal requirements is one of the elements that differ significantly ($p < 0.01$) in the ecology of the three plant communities (Table A1). Figure 3 confirms that microthermal species increase considerably with the increase in elevation (from A to C). The plant communities of the three sites studied present differences concerning L and N indexes (Table A1). From Figure 3, it is in fact possible to note how the site C community has more heliophilous species fitted to oligotrophic soils compared to the other sites and communities.

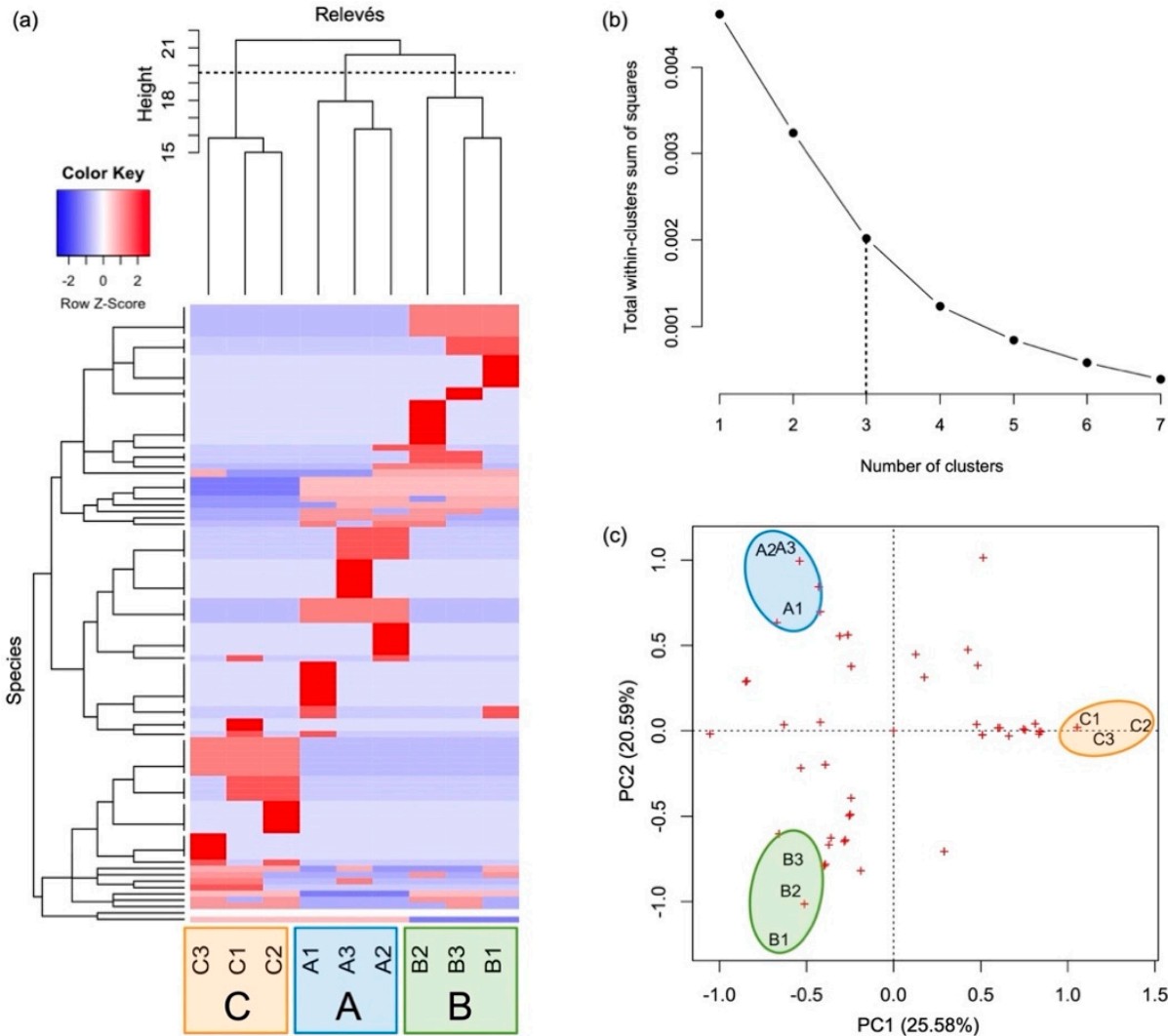

**Figure 2.** Hierarchical clustering heatmap and dendrogram of relevés (**a**) (capital letters indicate the sampling area where the relevés were carried out), distribution of total within-clusters sum of squares by number of relevé groups distinguished via hierarchical clustering (**b**) and PCA biplot of relevés (the red crosses are the species) (**c**).

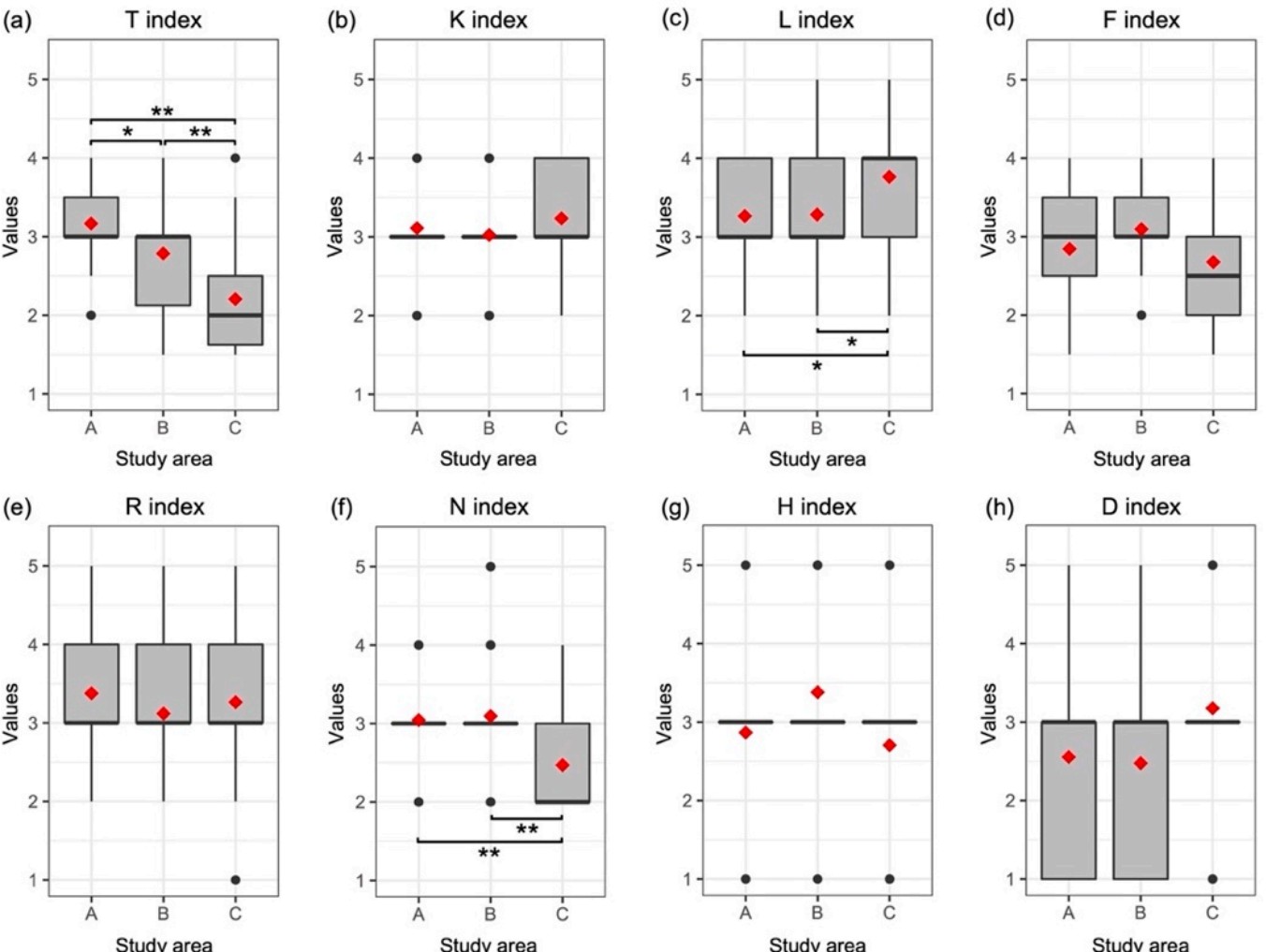

**Figure 3.** Ecological indexes of Landolt et al. [26]: Index of temperature—T (**a**); continentality—K (**b**); light intensity—L (**c**); soil moisture—F (**d**); substrate reaction—R (**e**); nutrients—N (**f**); humus—H (**g**); aeration—D (**h**). For each boxplot, the mean (red square) and median (black line) are highlighted. The asterisks show statistically significant differences: *, $p < 0.05$; **, $p < 0.01$.

### 3.2. Functional Strategy

Figure 4 shows the results of the analysis of the CSR functional strategy of *S. dodecandra* performed considering the three phenological stages: Pre-flowering (I), flowering (II), and post-flowering (III). *S. dedecandra* is an R/CR (ruderal/competitive-ruderal) species in each phenological stage and in each sampling area. Nevertheless, C, S, and R, values, considering each site, evidenced significant differences across phenological stages, excluding the C (competitor) index in the C site (Table A2). Figure 4 shows clear increases in the value of the S index from pre-flowering to post-flowering, a trend detectable in all the sampling areas. Considering C and R indexes, although there are differences among the various phenological stages, a clear trend of these parameters dependent on the phenological stage is not detectable (Figure 4). Considering all the values from the 180 leaf samples, the mean CSR strategy of *S. dodecandra* is R/CR: C = 25.14%, S = 8.61%, R = 66.25%.

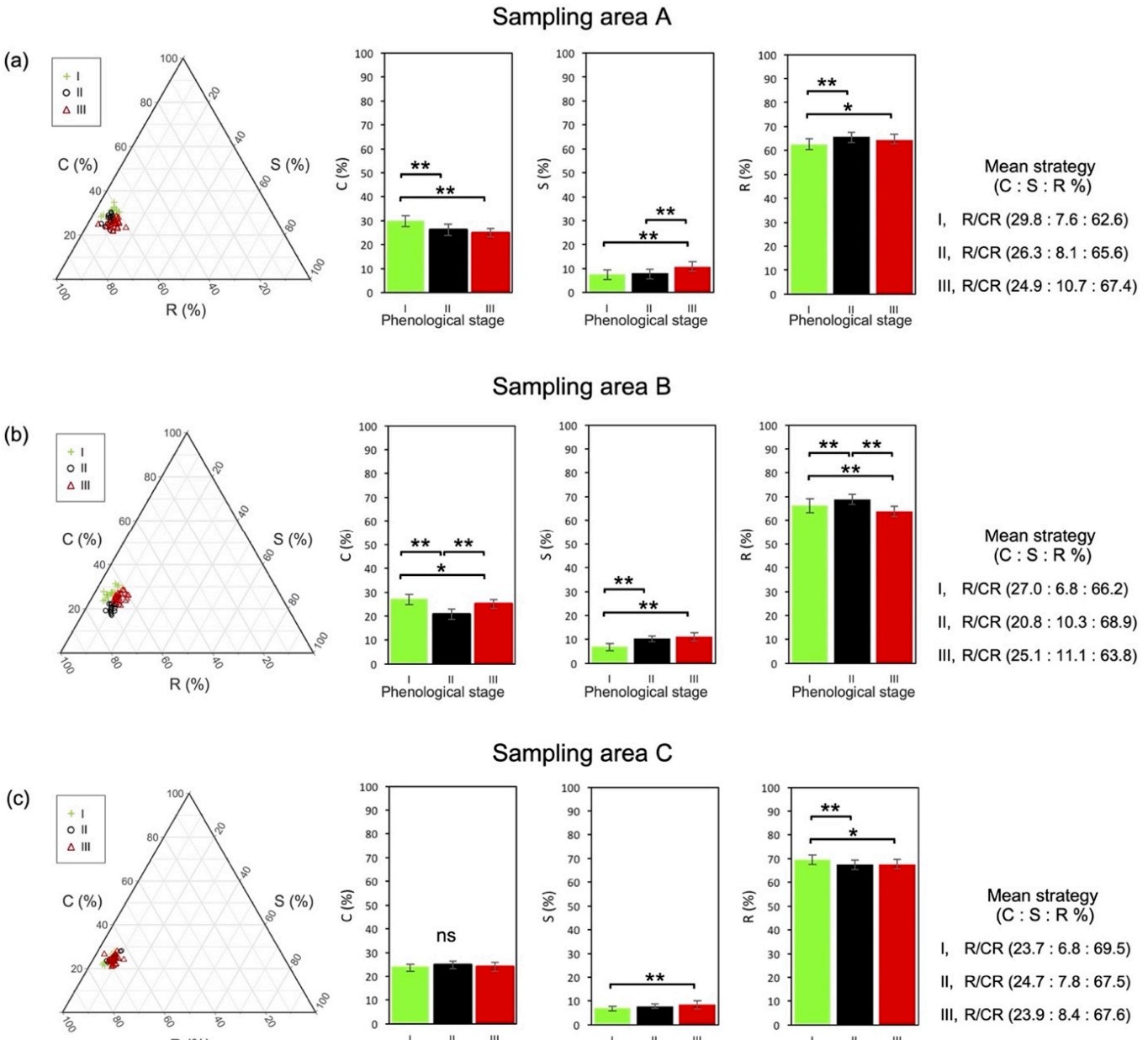

**Figure 4.** CSR triangular plots and histograms of the leaf samples of *S. dodecandra* collected in the three sampling areas (**a**–**c**) at different phenological stages: I, pre-flowering; II, flowering; III, post-flowering. The mean strategy was shown for each phenological stage of each sampling area. The asterisks indicate statistically significant differences: *, $p < 0.05$; **, $p < 0.01$; ns, not significant.

### 3.3. Bromatological Features

Table 2 shows the results of bromatological analysis (expressed as average $\pm$ SD) of each phenological stage, and the results of the one-way ANOVA. A great part of the dry weight of the samples of *S. dodecandra* analyzed is composed of NFC (47.01 $\pm$ 1.97%), NDF (33.75 $\pm$ 3.87%), and ADF (27.85 $\pm$ 3.32%). Only DM values, Ash, and PC of the three phenological stages are significantly different (Table 2). From pre-flowering to post-flowering, PC and ash content gradually decrease while the DM value increases. The average value of NDF, ADF and ADL increase from pre-flowering to post-flowering as well, although significant differences among the values of the three stages were not detectable (Table 2).

**Table 2.** Values returned by the bromatological analysis of the sample of *S. dodecandra* collected at different phenological stages (I, pre-flowering; II, flowering; III, post-flowering) and results of one-way ANOVA. Key: PC, protein content; NDF, neutral detergent fiber; ADF, acid detergent fiber; ADL, acid detergent lignin; EE, ether extract; NFC, non-fiber carbohydrates; DM, dry matter. The asterisks indicate statistically significant differences: *, $p < 0.05$; **, $p < 0.01$; ns, not significant.

| Parameter | Phenological Stage | | | One-Way ANOVA | | | |
|---|---|---|---|---|---|---|---|
| | I | II | III | Mean Square | $F_{2,6}$ | $p$ | |
| PC (% DW) | 12.92 ± 1.89 | 11.36 ± 0.13 | 8.73 ± 0.91 | 13.47 | 9.12 | 0.02 | * |
| NDF (% DW) | 31.33 ± 1.61 | 31.70 ± 3.88 | 38.22 ± 3.09 | 45.06 | 4.98 | 0.05 | ns |
| ADF (% DW) | 25.40 ± 1.64 | 26.52 ± 3.33 | 31.63 ± 3.22 | 33.15 | 4.13 | 0.07 | ns |
| ADL (% DW) | 4.36 ± 0.11 | 5.51 ± 1.52 | 7.123 ± 1.50 | 5.78 | 3.81 | 0.09 | ns |
| Ash (% DW) | 6.55 ± 0.25 | 5.99 ± 0.06 | 5.89 ± 0.26 | 0.39 | 8.69 | 0.02 | * |
| EE (% DW) | 2.08 ± 0.10 | 2.01 ± 0.21 | 2.18 ± 0.26 | 0.02 | 0.52 | 0.62 | ns |
| NFC (% DW) | 47.12 ± 3.62 | 48.94 ± 3.99 | 44.99 ± 3.62 | 11.71 | 0.83 | 0.48 | ns |
| DM (% FW) | 23.38 ± 2.06 | 27.95 ± 0.60 | 33.14 ± 1.73 | 71.64 | 28.40 | <0.01 | ** |

### 3.4. Pollen in Honey

The melissopalynological analysis identified 46 pollen types (Table S2) and defined the sample as sweet chestnut (*Castanea sativa*) honey, since the pollen of this tree was 70% ("dominant" pollen) of the total pollens (Figure 5, Table S2) The remaining 30% was mainly pollen of linden (*Tilia* spp.) and pollen of the species of *Pyrus* and/or *Sorbus* and *Rubus* genus, identified as "very frequent" pollen. *S. dodecandra* pollen is included in the "frequent" pollens, representing 9.5% of the "other pollen" (Figure 5) and, as such, it contributes to honey's aroma and taste.

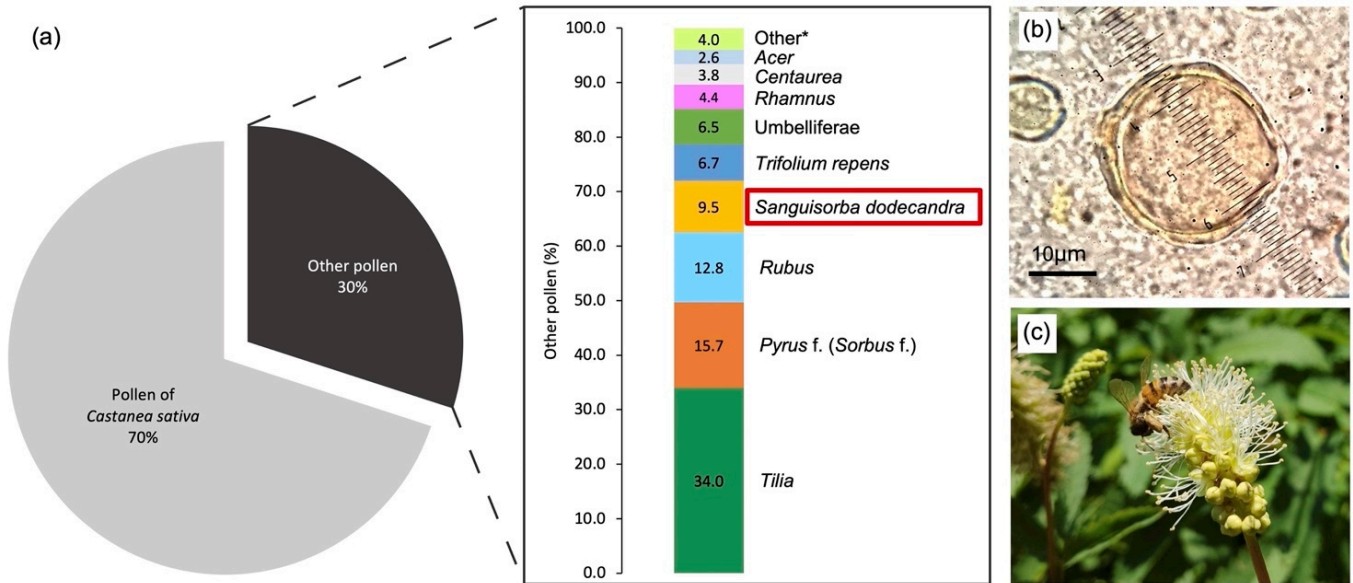

**Figure 5.** Type and percentage of pollen in the honey sample analyzed (*S. dodecandra* is highlighted in the red box) (**a**), microscopic images of pollen of *S. dodecandra* identified in the honey (**b**) and bee (*Apis mellifera*) on inflorescence of *S. dodecandra* (**c**). Key: *, sporadic pollen (<1%) of other 36 species/groups identified in the honey sample.

## 4. Discussion

### 4.1. Floristic and Ecological Features

The floristic and synecological analyses showed how *S. dodecandra* creates very similar plant communities considering the physiognomy (perhaps due to the dominance of this species) but very different from the floristic point of view. The multivariate analysis of

the relevés (Figure 2) evidenced several differences among the sampling areas. Such differences should be mainly due to the different elevations of the three areas analyzed, and then to the different temperatures. Species from the high mountain and sub-alpine belts were identified in site C, while they were not present at lower elevations (B and A), where more thermophilic species of the mountain belt were found. These data, supported by the synecological analysis (Figure 3), suggest and confirm the ecological plasticity of *S. dodecandra* in relation to the temperature, light intensity, and soil nutrients. *S. dodecandra*, in fact, is found either in open/sunny environments with nutrient-poor soils, such as the prairies over the trees limit (sampling area C), or at lower elevations where the presence of trees and/or shrubs limits sunlight and there are deeper and richer soils (sampling areas A and B).

The high ecological plasticity of *S. dedecandra* does not help in understanding the reason this species is confined in the Alps of Lombardy. The factor that likely limits its diffusion is not included in those considered in this study and by [26], and could be, as suggested by [10], the air moisture. In this author's opinion, in fact, *S. dodecandra* grows in areas with humid soil and/or high atmosphere humidity. This last condition, in areas with little water available in the soil, could be guaranteed by the fog. The high atmospheric humidity exigency could be justified by the fact that *S. dodecandra* is an amphistomatic species [10], a feature that has strong associations with the environment [50], and this could explain its confinement to the Orobic Alps (and surrounding areas), usually very humid compared to other areas of Lombardy and the Alps [2,51]. Studies comparing the air moisture of the habitats where *S. dodecandra* grows would be useful in clarifying this hypothesis.

Considering the CSR functional strategy, the results showed that *S. dodecandra* has an R/CR strategy in each sampling area and all phenological stages (Figure 4). This result is different from the result reported by [25,26], according to whom *S. dodecandra* could have a C/CSR (C = 64.84%, S = 16.57%, R = 18.59%) and "css", respectively. Following the opinion of these authors, *S. dodecandra* should be more of a competitor and less ruderal compared to the results of this study. Moreover, in [26], the CSR strategy was assigned empirically, and [25] did not specify the number of samples analyzed, their geographic origin, and their phenological stage. In [19], competitor species lived in eutrophic and low-disturbance environments. If *S. dodecandra* would really be a competitor, as indicated by [25,26], its capacity to tolerate disturbances, intended as all those phenomena able to partially or entirely destroy the plant biomass [19], would not be explicable. In our opinion, considering the different biotic and abiotic disturbances characterizing the environments where this plant grows (avalanche gullies, streams edges, pastures, and mowed grasslands), the R/CR strategy appears more correct, but different populations of *S. dodecandra* could have different functional strategies as evidenced by [20,22] for other endemic species of the Alps. This question could be clarified by studies on the intraspecific variability of the CSR strategy of *S. dodecandra*.

Although the functional strategy of *S. dodecandra* did not vary across the different phenological stages, the single values of C, S, and R showed significant variations, in particular considering S (Figure 4). This study analyzed the CSR strategy of *S. dodecandra* at different phenological stages for the first time, evidencing that this species becomes more of a stress-tolerator past the growing season (from pre-flowering to post-flowering). This could be explained by an increase in stresses (intended as abiotic limitations to productivity [19]) towards the end of the growing season, such as a shorter photoperiod and lower temperatures. The increase in S could be in relation to the increasing dry matter in the tissues of *S. dodecandra* as evidenced by the bromatological analysis (Table 2). The moisture content in the leaves is, in fact, one of the most-used parameters in the calculation of the CSR strategy [25] and, in the opinion of [19], stress-tolerator plants would be characterized by the hardness of foliar tissues.

### 4.2. Resource for Mountain Agro-Ecosystems

Bromatological analysis, other than evidencing that the epigean apparatus of *S. dodecandra* becomes harder (and then less palatable by cattle) passing the season, allowed the acquisition of information on the nutritional features of this plant. It was found that this species has a good protein content (approximately 12% of DW), especially in pre-flowering when its tissues are richer in minerals (ash content) and more tender (minor DM content) (Table 2). The protein content of *S. dodecandra* is similar to well-known good forage herbs found in alpine meadows and pastures, such as *Phleum alpinum* (PC = 11.51%), *Poa alpina* (PC = 11.46%), *Deschampsia cespitosa* (PC = 11.08%), *Anthoxanthum odoratum* (PC = 11.44%), and *Carex sempervirens* (PC = 12.31%) [52].

Other than the good protein content, *S. dodecandra* has a high content of NFC (44.99–48.94%), represented by starch and other reserve carbohydrates, making this plant a good fodder for dairy cows. NFC content necessary for this cattle category is, in fact, attested to by 36 to 44% of dry matter [53]. Protein and starch are fast energy resources, and the cheese-making properties of milk are dependent on energy supply [54,55] and other dietary factors [56]. Cheese-making properties are important for both the quality and yield of cheese [57]. In general, high-forage diets may help to increase the concentrations of some important fatty acids in cow milk [58] and diets rich in herbs seem to increase the recovery of dietary α-linolenic acid (18:3n−3) in milk [55,59], an attribute where phenolic compounds might be involved [60], and specific secondary plant metabolites have already been discussed as possible causes for the high 18:3n−3 concentrations in alpine milk [55]. No metabolomic investigation has been conducted on *S. dodecandra* to date, although we have diverse studies on the more common species *S. officinalis*, since it is a known ethnobotanical remedy, for example, in Chinese popular medicine, and is noticeably rich in phenolic compounds [61,62]. A further field of investigation would be to explore the phytochemical profile of this endemic species and understand whether the popular knowledge on its properties (increase fat content in dairy milk) is verifiable or if it contains interesting molecules for herbal medicine such as *S. officinalis*.

*S. dodecandra* never received a forage index (FI) to evaluate its forage value in meadows and pastures [35,63–67]. The FI, proposed by Delpech in [68], is a synthetic index since it considers different aspects such as the nutritional value, palatability, and grazing tolerance of each species. Based on the present study's results and comparing the FI of species comparable to *S. dodecandra* in terms of ecological/bromatological features [63,64], it is possible to suggest an FI = 7 for this species. This value refers to the Klapp–Stahlin scale [63] where FI varies from −1 (refused or toxic species) to 8 (highest preference). The FI value proposed for *S. dodecandra* could be useful for agronomists and botanists for the correct assessment of the forage value of Lombardy meadows and pastures where this species is present.

Besides being an interesting fodder plant, the melissopalynological analysis evidenced that *S. dodecandra* is also important for beekeeping. The honey produced near the sampling sites is a good sign that this plant pollen was detected (Figure 5). *S. dodecandra* pollen is more easily identifiable compared to other Rosaceae [69] and other species of the *Sanguisorba* genus [8,70]. From these results, we can affirm that the honeybee is certainly among *S. dodecandra* pollinators. Honeybees from the same colony forage across areas spanning up to several hundred square kilometers, and at linear distances as far as 9 km from the hive [71], and although honeybees are considered supergeneralists in their foraging choices, there are certain key species or plant groups that are particularly important in honeybee foraging [72], and *S. dodecandra* could be among them. Some of these particularly important groups of plants are species of the Rosaceae family, and some broad-leaved trees such as chestnut (*Castanea sativa*) or plants of *Tilia* genus, which were found also in our sample. The metabolites of *C. sativa* found to be the "dominant" pollen in this research seem to be correlated to individual and collective honeybee behavior driven by proximate physiological mechanisms that involve the tryptophan metabolism via the kynurenine pathway [37]. In this framework, plant–insect communication, and which pollinator a plant

target uses for its reproductive strategy, is a fascinating field to explore [73] in the husbandry field of beekeeping. Although chestnut resulted in being the "dominant" pollen in our sample, the fact that this pollen is classified as "overrepresented" in honey [74] should be considered, and the importance of *S. dodecandra* as a nectar source could be greater compared to its pollen presence. The nectar potential and the preference of honeybees could relate to the availability and abundance of the plant, the quality and abundance of the nectar and pollen, and/or specific nutrients or elements offered by the species considered or even neurological aspects, as mentioned above, and more should be investigated about this endemic plant.

An in-depth characterization of the secondary metabolite profile of flowers was realized for *S. officinalis* by Bunse et al. in [62], describing fresh flowers of this plant as characterized by a noticeable odor reminiscent of amines, and the investigation was then focused on this compound class. One major compound was identified as 2-phenylethylamine (PEA), the precursor compound of all other phenylethylamine alkaloids, and it was assumed that PEA likely plays an important role in attracting pollinators, and identified in Diptera as the most important insect order of *S. officinalis* pollinators. *S. dodecandra* flower scent is, instead, described to be similar to cyclamen/linden, and linden honey is described as "floral, fresh and citrusy, medicinal, balsamic, very persistent" (Gianoncelli C., personal communication). To date, specific pollinators of *S. dodecandra* are not known, and this would be an interesting field of further investigation about this plant.

Based on the results of this research, a remarkable research field in the future could be to evaluate if *S. dodecandra* is characterized by specific nutritional and/or phytochemical features transferable to honeybee and dairy products. These aspects would be important to valorize agri-food products and related sustainable value chains linked to this plant and the mountain areas where it grows.

As previously reasoned, dairy production and honey are strongly linked to the area where they are produced, with distinct botanical features. Their uniqueness could then represent an opportunity for sustainable local resource use in mountain marginal areas through added-value and exclusive-quality productions.

## 5. Conclusions

This multidisciplinary study allowed a deeper knowledge of *S. dodecandra* ecology and showed that this species is a resource for sustainable pastoralism and honey production in the Alps of the Lombardy region where this species grows.

The synecological analysis showed that this species is euriecious and able to form plant communities physiognomically similar but floristically different, based mainly on the different elevations. The definition of CSR functional strategy confirmed the tolerance of this species to disturbances (biotic and abiotic) characterizing the environments where it grows: Instability of the debris/soil substrate (for example, at creek edges and avalanche gullies) or mowing/grazing (for instance, in grazed/mowed grasslands).

The bromatological analysis confirmed the ethnobotanical knowledge of the good forage value of this plant and allowed us to attribute it an FI value. The FI will be useful for agronomists/botanists and land managers to better evaluate the forage value of Lombardy meadows and pastures, and to valorize this mountain agro-eosystem.

The results of the melissopalynological analysis also highlighted the importance of the role of *S. dodecandra* for the (agro-)ecosystems of the study area. In fact, it is attractive for honeybees, which can now be included among its pollinators and that allow the production of a unique honey due to the good presence of pollen of this endemic plant. The forage value, ecological characteristics, and importance of *S. dodecandra* in the production of honey are aspects that make this species very interesting (compared to most of the endemic species of the Alps) for the enhancement of the mountain agro-ecosystems of the Lombardy Alps and their agri-food products.

This research is an example of how the study of local biological resources (such as endemic plants) can be useful for identifying "new" agricultural resources useful for the

sustainable development of mountain areas, rather than for purely scientific and protection purposes, with the sustainable use of biodiversity also useful for its conservation.

**Supplementary Materials:** The following supporting information can be downloaded at: https://www.mdpi.com/article/10.3390/su14116825/s1, Table S1: Table of relevés. Cover indices refer to the Braun–Blanquet abundance/dominance scale [41]: r, rare; +, <1%; 1, 1–5%; 2, 6–25%; 3, 26–50%; 4, 51–75%; 5, 76–100%; Table S2: Relative frequencies of the main pollen types in the honey sample.

**Author Contributions:** Conceptualization, L.G. and V.L.; methodology, L.G., V.L., C.G., and A.T.; software, L.G.; validation, L.G., A.T., and A.G.; formal analysis, L.G., V.L., C.G., and A.T.; investigation, L.G. and V.L.; data curation, L.G., V.L., and A.T.; writing—original draft preparation, L.G. and V.L.; writing—review and editing, L.G. and V.L.; supervision, L.G., A.T., and A.G.; project administration, A.G.; funding acquisition, L.G. and A.G. All authors have read and agreed to the published version of the manuscript.

**Funding:** This research was supported by the "Montagne: Living Labs di innovazione per la transizione ecologica e digitale" project and the PhD school of Environmental Sciences (ES) of University of Milan.

**Institutional Review Board Statement:** Not applicable.

**Informed Consent Statement:** Not applicable.

**Acknowledgments:** We would like to thank Federico Caccia and Alessia Rodari for the help provided in the field and in the laboratory.

**Conflicts of Interest:** The authors declare no conflict of interest.

## Appendix A

**Table A1.** One-way ANOVA results of sampling area effect on ecological indices of Landolt et al. [26]: T, index of temperature; K, index of continentality; L, index of light intensity; F, soil moisture index; R, index of substrate reaction; N, index of nutrients; H, index of humus; D, aeration index. The asterisks show statistically significant differences: *, $p < 0.05$; **, $p < 0.01$; ns, not significant.

| Source of Variance | Mean Square | $F_{6,118}$ | $p$ | |
|---|---|---|---|---|
| T | 8.95 | 24.22 | <0.01 | ** |
| K | 0.42 | 1.32 | 1.00 | ns |
| L | 2.29 | 6.00 | 0.03 | * |
| F | 1.70 | 4.85 | 0.08 | ns |
| R | 0.73 | 1.17 | 1.00 | ns |
| N | 4.40 | 9.28 | <0.01 | ** |
| H | 4.92 | 5.27 | 0.05 | ns |
| D | 5.38 | 3.72 | 0.22 | ns |

**Table A2.** One-way ANOVA results of phenological stage effect (pre-flowering, flowering, and post-flowering) on C, S, and R values. Key: **, $p < 0.01$; ns, not significant.

| Sampling Area | Source of Variance | Mean Square | $F_{2,57}$ | $p$ | |
|---|---|---|---|---|---|
| | C | 124.41 | 27.40 | <0.01 | ** |
| A | S | 53.06 | 14.30 | <0.01 | ** |
| | R | 44.94 | 8.88 | <0.01 | ** |
| | C | 200.43 | 47.92 | <0.01 | ** |
| B | S | 103.52 | 50.38 | <0.01 | ** |
| | R | 127.79 | 22.28 | <0.01 | ** |
| | C | 6.61 | 2.39 | 0.10 | ns |
| C | S | 13.26 | 7.80 | <0.01 | ** |
| | R | 26.33 | 6.58 | <0.01 | ** |

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
