# Peer review of "Endemic Plants Can Be Resources for Mountain Agro-Ecosystems: The Case of Sanguisorba dodecandra Moretti"

_sustainability, doi:10.3390/su14116825_

Round 1

Reviewer 1 Report

The manuscript "Endemic plants can be resources for mountain agro-ecosystems: the case of Sanguisorba dodecandra Moretti" is a very interesting text that presents a peculiar feature of a specific area of the Italian country. It think that the manuscript is well written and it describes in a very specific way all the analyses performed. In particular, all the analyses presented the right number of biological and technical replicates and the statistical analyses are well described. At the end of the reading I was curious to know the phytochemical profile of this endemic species and evaluate its correlation with the cheese-making properties of the milk. I find in the text only few msistakes such as: the authors missed a space in line 193 between 12 and h; while in line 378 the authors forgot the closure of brackets.

Reviewer 2 Report

The paper is based on sound scientific procedures, from field work to data analysis, although I would classify it as more of a case study with local interest.

It is well written and reasonably succinct for the amount of work it involves. It would, however, greatly benefit from English language improvement. I found it difficult to understand in many places. A few examples are shown below but there are many more. Since I am not an expert, and certainly not a native, English speaker myself, I am not in a position to correct the text but correction is required.

Line 69: “… Grime theory was applied …” should be “… Grime’s theory was applied …”

Line 84 – 85: “… through added  value and unique quality productions [27]. “ perhaps should be  “… through added  value and unique quality products [27]. “

Line 91 – 92: “… cheeses … resulted influenced by the different predominant plants of grasslands [31].” Two verbs in one sentence.

Line 154: “In each sampling areas (A, B and C) … “ should be “In each sampling area (A, B and C) …“

Line 243 – 244: “The most of them result perennial herbaceous plants …” should be “Most of them were found to be herbaceous perennials …”

Line 336: “(B e A)” should be “(B and A)”

Line 358: “… in which opinion S. dodecandra could …” may be should be “… in whose opinion [or: according to whom] S. dodecandra could …”

Reviewer 3 Report

Review for sustainability-1738569

The study is interesting study on ecological features of S. dodecandra and its importance for pastoralism and honey production and it provides some useful practical details. The authors explained their aims, methods, and results but some sentences of the paper manuscript are not easy to understand. Therefore, I would like to make some suggestions to improve the quality of the paper as below:

In general, some parts of the manuscript are not easy understand. There are many long sentences, and this situation disrupts the flow of the subject and the continuity of the reading. Because of this reason, authors should re-reconsider to write the manuscript with short sentences.

Introduction

Lines 36-40: Please re-phase this paragraph with shorter separate sentences.

Lines 46-48: Please re-phase this sentence.

Lines 51-52: According to the IUCN Red List of threatened species, this species in fact belongs to the category “Near Threatened – NT” -> According to the IUCN Red List of threatened species, this species considered as “Near Threatened – NT”. Because of this reason, the species has a conservation concern.

Line 56: various authors believe that…. -> different studies stated that…

Line 62: Additionally, there are no data on the synecological features -> Moreover, we did not find any data on synecological features.

Lines 64-75: This part of the paper should be rewritten since it is not easy to understand.

Line 65-66 “This last analysis is based on Grime theory” what do you mean with “this last analysis”?

Line 113: I think such sentence would be better to explain the aim of the study: “In this study, therefore, we aimed to understand the ecological features of S. dodecandra which is poorly known. In this context, its role in some mountain agricultural activities as pastoralism and honey production were investigated….”

Materials and Methods

Line 146: “Sampling areas” is enough for title of the Table 1.

Line 165: carried out  -> performed

Line 168: A reference is needed.

The R codes used for the analyses may be given in the supplementary information to increase the reproducibility

Line 182: carried out  -> performed

Discussion

Line 423: interesting for beekeeping -> important for also beekeeping

Line 423: a good presence-> is a good sign

Line 458-460: Please rewrite with shorter sentences.

Line 463-466: Please rewrite this paragraph with separate sentences.

Reviewer 4 Report

This manuscript has an interesting thesis and is well written. Discussion is complex.

I have only minor comments:

1. Did you collect just one honey sample from the collection year?

2. Were the bees' hives stationery or transferred around the area during the season. Provide this information in M&M.

3. What was the criterion for choosing the area for phytosociological releves. How distant were those areas, and were they uniform?
